# Portable Device for Quick Detection of Viable Bacteria in Water

**DOI:** 10.3390/mi11121079

**Published:** 2020-12-04

**Authors:** Yu-Hsiang Liao, Karthickraj Muthuramalingam, Kuo-Hao Tung, Ho-Hsien Chuan, Ko-Yuan Liang, Chen-Peng Hsu, Chao-Min Cheng

**Affiliations:** 1Institute of Biomedical Engineering, National Tsing Hua University, Hsinchu 300, Taiwan; a29248510@hotmail.com (Y.-H.L.); t19872@hotmail.com (K.-H.T.); 2Electronic and Optoelectronic System Research Laboratories, Industrial Technology Research Institute, Hsinchu 310, Taiwan; karthickraj@itri.org.tw; 3Department of Surgery, National Taiwan University Hospital, Chu-Tung Branch, Hsinchu 300, Taiwan; b91401091@ntu.edu.tw; 4Center for Environmental Toxin and Emerging-Contaminant Research, Cheng Shiu University, Kaohsiung 833, Taiwan; kuyuan68@gmail.com; 5Super Micro Mass Research and Technology Center, Cheng Shiu University, Kaohsiung 833, Taiwan

**Keywords:** bacterial detection, portable device, MTT, PMS, colorimetry

## Abstract

(1) Background: Access to clean water is a very important factor for human life. However, pathogenic microorganisms in drinking water often cause diseases, and convenient/inexpensive testing methods are urgently needed. (2) Methods: The reagent contains 3-(4,5-dimethylthiazol-2-yl)-2,5-diphenyltetrazolium bromide (MTT) and phenazine methosulfate (PMS) and can react with succinate dehydrogenase within bacterial cell membranes to produce visible purple crystals. The colorimetric change of the reagent after reaction can be measured by a sensor (AS7262). (3) Results: Compared with traditional methods, our device is simple to operate and can provide rapid (i.e., 5 min) semi-quantitative results regarding the concentration of bacteria within a test sample. (4) Conclusions: This easy-to-use device, which employs MTT-PMS reagents, can be regarded as a potential and portable tool for rapid water quality determination.

## 1. Introduction

Water is an indispensable natural resource for sustaining human life. Clean water is an important factor for economic development and food production [1]. However, there are approximately 2.1 billion people in the world without safe drinking water [2]. Because pathogenic microorganisms in drinking water frequently affect human health, it is necessary to develop a device for easy and rapid monitoring of water quality. The indicator bacteria commonly used to monitor water quality are coliforms, *Escherichia coli* and Pseudomonas aeruginosa [3]. *E. coli* is a member of the fecal coliform group and is considered to be an important water quality detection indicator [4]. Food and water are frequently contaminated with *E. coli* from human or animal waste that can survive for a long time in a changing environment [5]. For all of these reasons, measurement of *E. coli* content in water is both a viable and extremely valuable approach for determining water quality. 

Existing bacterial detection techniques are primarily divided into two major categories: (1) traditional culture methods and (2) biotechnological approaches to distinguish the presence of *E. coli* [6]. The traditional culture method involves colony counting of diluted or whole sample solutions to determine bacterial concentration [7]. The disadvantage of this method is that it takes a long time to obtain results (24–48 h), and it requires an incubator. Biotechnological approaches include polymerase chain reaction amplification (PCR), fluorescence analysis, and traditional enzyme-linked immunoassay (ELISA). However, the above methods have many disadvantages including the following: (1) cost of materials (enzymes, antibodies, and antigens), (2) low-temperature storage requirements, (3) skilled and experienced operational needs, (4) different experimental methods must be combined with different instruments; and (5) long and complicated operation methodology [6,8,9]. If people in developing countries want to confirm whether their water resources are polluted, they must consider cost, convenience, and whether professional training is required for analysis. These multiple shortcomings make it difficult for people to determine water safety on their own. 

A variety of *E. coli* detection technology currently exists including each of the following: (1) a capacitive matrix that leverages the matrix topology of polycolonal antibodies on the sensor surface to detect *E. coli* via capacitance changes at concentrations as low as 10^3^ CFU/mL [10]; (2) a microfluidic biosensor device that combines gold nanoparticles with immunodetection methodology and smartphone imaging to rapidly (within 1 h), easily (in the field), and inexpensively detect *E. coli* at concentrations as low as 50 CFU/mL [11]; and (3) a handheld heating device to carry out loop-mediated isothermal amplification (LAMP) on paper in order to produce a colorimetric paper-based biosensor that can be read by the naked eye. Using this approach, nucleic acids can be extracted, amplified, and detected within 1 h, with an LOD of 10–10^3^ CFU/mL [12]. CFU means that each bacterial cell grows on the culture medium to form a single colony that is easy to distinguish. Each colony unit is called 1 CFU.

In this study, we leveraged the reaction of 3-(4,5-Dimethylthiazol-2-yl)-2,5-diphenyltetrazolium bromide (MTT) and phenazine methosulfate (PMS) when exposed to succinate dehydrogenase within bacterial cell membranes. Here, MTT reacts with succinate dehydrogenase and cytochrome C to cleave the tetrazole heterocyclic to form purple formazan crystals, while PMS acts as an intermediate electron acceptor to assist in the above reduction [13,14]. We added TE buffer (Tris-Ethylenediaminetetraacetic acid, Tris-EDTA) to help break the bacterial cell wall and facilitate the entry of the reagent [15]. At the end of the reaction, we added sodium hydroxide as a signal amplification reagent. Sodium hydroxide can accelerate the reduction of MTT to purple formazan crystals, which makes the resulting optical signal more easily visible [16]. Our colorimetric device point-of-care technology can be implemented and read at any site (e.g., riverside, drainage ditch) where bacteria inspection is required. 

## 2. Materials and Methods 

### 2.1. Reagents 

Thiazolyl blue tetrazolium bromide (Sigma Aldrich, St. Louis, MO, USA); phenazine methosulfate (Sigma Aldrich, St. Louis, MO, USA); sodium hydroxide (Showa Chemical, Tokyo, Japan); Tris-EDTA buffer (Genemark, Taipei, Taiwan). 

### 2.2. Bacterial Colorimetric Device Design 

This device can be divided into four parts: (1) light source, (2) cuvette holder, (3) sensor (AS7262: ams, Unterpremstätten, Austria), and (4) microcontroller (Arduino Nano V3.0: Arduino, Somerville, MA, US). We used a white LED to illuminate the sample in the cuvette holder. Light that was not absorbed by the sample was scattered or reflected to our six-channel colorimetric device, which was capable of capturing result signals at a variety of wavelengths, i.e., 450, 500, 550, 570, 600, and 650 nm respectively. The transmissive optical setup and device image see Figure 1.

### 2.3. Colorimetric Assays 

Our process was divided into 4 steps. First, we used TE buffer solution for 5 min to weaken the outer membrane of the cell and increase permeability. The second step was the addition of MTT-PMS reagent to react with the succinate dehydrogenase within the bacterial membrane. During this reaction, the reagent turned into a purple crystal solution (Figure 2). During this reaction, succinate dehydrogenase acts upon the bacterial electron transport chain to cleave the tetrazolium on the MTT molecule, which reduces the MTT to purple crystals, and the PMS acts as an intermediate electron acceptor to assist the reduction. The third of four steps was the addition of sodium hydroxide as a signal amplification reagent. Sodium hydroxide can accelerate the process of reducing MTT to purple crystals, to provide a more visibly obvious colorimetric change (Figure 2). In the fourth and final step, we placed the cuvette into the bacterial colorimetric device to measure the colorimetric change and allowed it to run for 5 min. Based on our experimental results, we observed that, as the bacterial concentration in the solution increased, the colorimetric of the reagent changes were more notable, and the rate of colorimetric change was more rapid. 

## 3. Results and Discussion 

### 3.1. Feasibility Analysis for Bacterial Reagent 

Based on our previous study [17,18], we chose to employ an MTT-PMS reagent mechanism to produce colorimetric differences to gauge bacterial concentration in a water sample. This mechanism is stable and significantly reduces bacterial detection time from several hours to 5 min. We deviated from the abovementioned study with regard to reagent application and reagent formulation and added EDTA to weaken the outer membrane to facilitate reagent penetration and speed up reaction time. The reagent reacted with the bacteria in the sample to produce a notable and readable colorimetric change within 5 min. We observed that reagent absorption and the resulting colorimetric change of samples was positively correlated with sample bacterial concentration and reaction time. As concentration rose, the colorimetric change was more intense. The longer the reaction was allowed to take place, the greater the colorimetric change. We selected 5 min as a suitable test point. 

According to the above feasibility analysis, we determined the optimal experimental parameters and conditions. Based on these conditions, we measured the change in absorbance of the reagent and bacteria at a wavelength of 595 nm for 5 min and validated the estimated number of bacteria in the sample using the spread-plate method. Figure 3 shows that the bacterial concentration in our samples ranged from 2–10^8^ CFU/mL. The absorbance values are shown in Figure 3 as the average of multiple experimental results (*n* = 8). Based on the experimental results, we observed significant differences in absorbance values between different amounts of bacteria. This method for determining water quality thus demonstrates stability and reproducibility. We believe that our reagent-based approach is feasible and effective for rapid detection of bacterial concentration in water. 

### 3.2. Colorimetric Devicecalibration 

In this experiment, bacteria sample colorimetric depth was recorded within 5 min using a colorimetric device and the data were transferred to a computer via a microcontroller. The colorimetric depth is dependent upon bacterial concentration in the sample. Figure 4a and Figure 5a show the colorimetric depths of six different concentrations measured continuously over 5 min. The unit of colorimetric value is arbitrary. Figure 4b and Figure 5b show the calibration model that was used to relate sample colorimetric depth to bacterial concentration at 5 min. This calibration model was used to rapidly semi-quantify the bacterial concentration in unknown water samples. Because *E. coli* is a primary indicator of water pollution, we used it here to establish a prediction model. Note the reagent used in this method is capable of counting active and respiring cells. 

As shown in Figure 4b and Figure 5b, the prediction model in PBS and water uses Hill’s equation model. Three times the standard deviation of the blank test intensity was 332.8665 (*n* = 8) and 261.75564 (*n* = 8) in PBS and water, respectively, which was considered the limit of detection (LOD). Fitting that back into Hill’s equation, the LOD for bacterial concentration was determined to be 3.43 × 10^5^ and 1.41 × 10^6^ CFU/mL, respectively. 

The U.S. Environmental Protection Agency (EPA) standard for tap water is 5 × 10^2^ CFU/mL, and our current technology can measure values as low as approximately 10^5^–10^6^ CFU/mL (Figure 6). Although it cannot reach the concentration level required by the standard value, we provided a rapid detection method that can quickly eliminate unsuitable water sources. Future optimization of the reagent formulation will improve the detection concentration level to match the standard value. 

## 4. Discussion 

Based on our detection results, this proof-of-concept study supports the idea that this approach can be used to rapidly obtain bacterial concentrations in water and provide results similar to those available via standard laboratory procedures (in PBS). Further, the colorimetric device reader employed can provide similar results to those of a more expensive microplate reader. Compared with traditional methods, as shown in Table 1, this operation is simple, and the assay can be completed in 5 min. This method can be used to rapidly semi-quantify bacterial concentration in water by leveraging a simple reagent colorimetric change. 

The estimated cost of our device is only US $12–17. The main reason for the low cost is the use of cost-effective spectral sensors and an integrated LED driver with a programmable current, which helps simplify device design. The following is the cost breakdown: (1) white LED, $0.02; (2) AS7262, $5.3; (3) control unit, $3; (4) mechanics and cover, $3; (5) battery, $1; and (6) reagent cost for one test, $5. 

Bacterial inhibition of GOX (glucose oxidase)-catalyzed reaction (BIGR) has been used to quantify sample bacterial concentration as low as 10^4^ CFU/mL in 20 min by consuming glucose produced as a result of bacterial respiration [26]. The use of enzyme–nanoparticle assemblies provides a colorimetric sensing approach that can determine 10^2^ CFU/mL in a solution within a few minutes and can provide a naked-eye, visible result for samples containing 10^4^ CFU/mL [27]. Another point of care (POC) device uses isothermal amplifications, hydroxy naphthol blue (HNB) dye, and calcein dye to detect *E. coli* and *S. aureus*, respectively. This device can detect 30 CFU/mL *E. coli* and 2 × 10^2^ CFU/mL *S. aureus* within one hour, but a specific primer is required [28]. By linking D-amino acid substrate with magnetic nanoparticles, a colorimetric method for detecting Listeria monocytogenes was developed. The LOD of this sensor was 2.17 × 10^2^ CFU/mL [29]. Regarding the five methods for colorimetric bacteria detection mentioned above, we believe that we can reduce the LOD of our device by employing reagent optimization. 

Note that while MTT formazan is composed of purple insoluble crystals, the entire solution turns dark green after 5 min of reaction time because PMS passes through the semiquinone intermediate during the reduction process [30], causing a resultant colorimetric change from yellow to green. 

The standard for total bacterial count for fresh chilled fish in Egypt is 10^6^ CFU/mL [31]. Bacteriological analysis of retail fish samples indicated that the total bacterial count in tilapia fish samples on the market was greater than 10^5^ CFU/mL [32]. Two kinds of pathogenic bacteria *Ps. fluorescens* and *A. hydrophila* bacteria were detected in the water samples from two water rearing tanks via culture-dependent methods and culture-independent methods. The culture-dependent method indicated a bacterial concentration of less than 10^3^ CFU/mL, and the culture-independent method indicated a bacterial concentration of greater than 10^6^ CFU/mL. This indicates that both culture-dependent and culture-independent methods should be used to evaluate the bacterial stability of aquaculture [33]. Water resources in developing countries (e.g., Congo and India) may be polluted by the discharge of urine and feces [34,35,36]. For detecting the bacterial count of bacteria in wastewater or in the terrarium, it is important to provide an early warning. The device we have developed and described here would be highly useful for such efforts. 

The colorimetric device employed in the sample reader is small in size and facilitates the production of a portable device that can be used to measure samples immediately after sampling. Based on the above advantages, this method can be used for real-time monitoring of groundwater and river water quality. Real-time monitoring of water quality can prevent people from drinking unclean water while waiting for more complete water analysis. In terms of fast detection, simple operation, and portability, this device can be regarded as a potential water quality detection tool.

## Figures and Tables

**Figure 1 micromachines-11-01079-f001:**
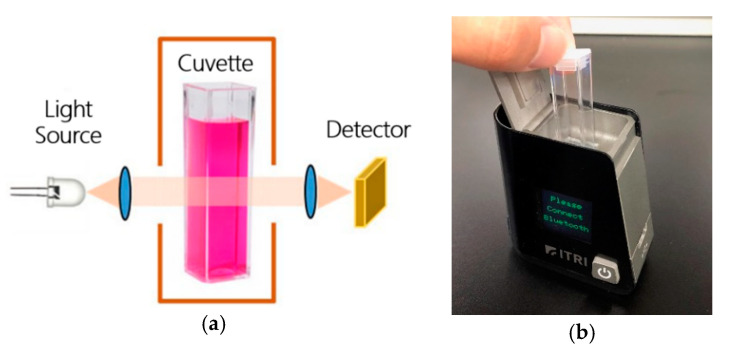
Bacterial colorimetric device module design and development: (**a**) transmissive optical setup schematic and (**b**) a real image of the device.

**Figure 2 micromachines-11-01079-f002:**
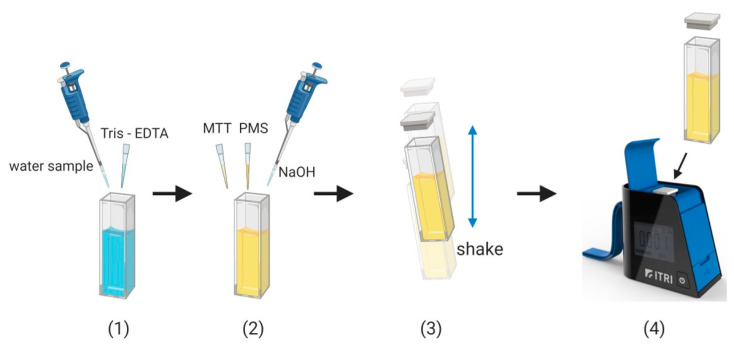
Step-by-step procedure chart for bacteria detection. (**1**) Add water sample and Tris-EDTA and incubate for five minutes. (**2**) Add the bacteria detection reagent and then quickly close the lid. (**3**) Shake for 3 s to mix the reagents quickly. (**4**) Put the cuvette into the colorimetric device and take measurements. Created with BioRender.com. (https://app.biorender.com/).

**Figure 3 micromachines-11-01079-f003:**
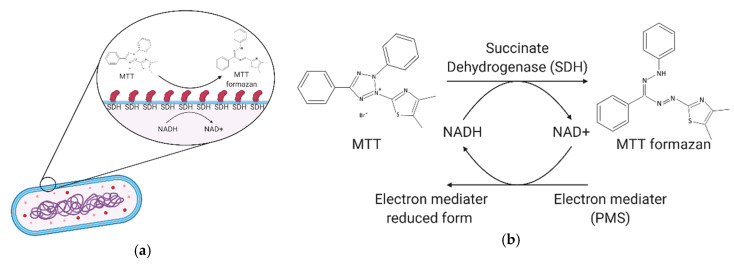
Mechanism of the bacterial reagent assay: (**a**) schematic diagram of the bacteria reagent assay and (**b**) MTT-PMS mechanism of our bacterial analytical device. Created with BioRender.com.

**Figure 4 micromachines-11-01079-f004:**
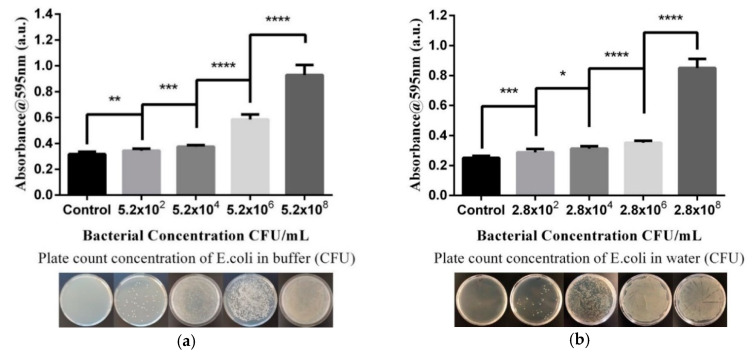
Absorbance value after reaction of bacterial amount and reagent (*n* = 8), with the vertical axis indicating the mean intensity value of absorbance and the horizontal axis indicating concentration of bacteria in (**a**) PBS and (**b**) water. The data are measured using a Tecan sunrise ELISA reader. (* *p* < 0.05, ** *p* < 0.01, *** *p* < 0.001, **** *p* < 0.0001).

**Figure 5 micromachines-11-01079-f005:**
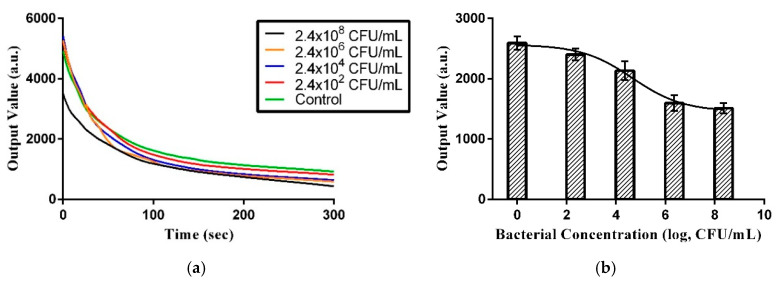
(**a**) Measurement changes over time demonstrating the output values of bacterial samples of different concentrations. (**b**) Prediction model for bacterial concentration using PBS and 5 min colorimetric values.

**Figure 6 micromachines-11-01079-f006:**
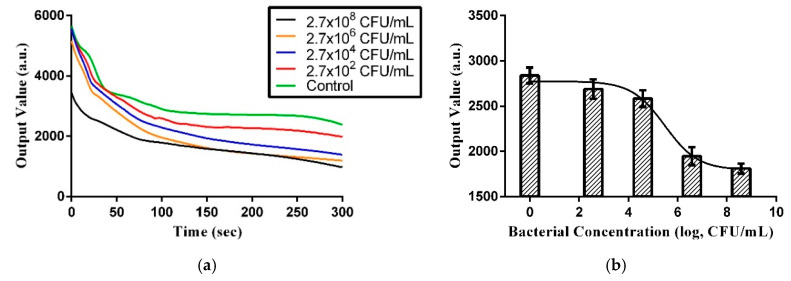
(**a**) Measurement changes over time demonstrating the output values of bacterial samples at different concentrations. (**b**) Prediction model for bacterial concentration using water and 5 min colorimetric values.

**Table 1 micromachines-11-01079-t001:** Comparison of current detection technologies.

Method	Time	Cost	LOD
Multi-tube fermentation	A few days	Low	<3–10^3^ MPN/100 mL
spread-plate	A few days	Low	30 CFU/cm^2^ [19]
q-PCR	A few hours	High	10^2^–10^4^ cells/gram [20,21] units and tens cells/mL [22]
ELISA	A few hours	High	10^3^–10^5^ CFU/mL [23,24,25] with a few hours of enrichment <1 CFU/gram [24,25]
Portable device with bacteria detection reagent	<15 min	Low	10^5^–10^6^ CFU/mL

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
