# Peer review of "Portable Device for Quick Detection of Viable Bacteria in Water"

_micromachines, 2020, doi:10.3390/mi11121079_

Round 1

Reviewer 1 Report

The paper presents a colorimetric detection scheme and device for bacterial detection in drinking water. The method has some novelty that would support its publication. However, there are two major weak points: the performance of the device is not properly compared with the state of the art and the current LOD of the device would not permit its direct use for the intended application. (see details in the comments below). My major and minor comments:

  • All acronyms should be extended even in the abstract (MTT, PMS).
  • There are some typos in the text (e.g. “regarding the concentration of bacterial” in the abstract), please carefully check the text of the manuscript.
  • I suggest removing Tris-EDTA from the Keywords. I suggest colorimetry instead.
  • The caption of Fig. 1 is wrong for many reasons. a) Is not about the device structure, this is merely an illustration about the colorimetric sensing – a transmissive optical setup. b) is also not the “schematic” of the device, this is either a photograph or a 3D rendered design.
  • Also, the terms “bacterial color sensing” or “bacterial color device” is not correct. The device does not measure the color of bacteria, but the color change of an indicator. This is a standard colorimetric technique, please refer to it as such. Use “colorimetric sensing” “colorimetric sensor/device” etc.
  • “The longer the reaction was allowed to take place, the greater the color change. We selected 10 minutes as a suitable test point.” What is this decision is based on? Since Fig. 4 and 5 only present the curves for the first 5 minutes, signal stability cannot be judged.
  • In Table 1 the LOD for q-PCR (10^5-10^6 CFU/mL) is an example from a specific paper for the enumeration of Streptococcus mutants from oral samples. Generally, the LOD of q-PCR can be orders of magnitude better. Please try to give a more relevant comparison.

https://www.frontiersin.org/articles/10.3389/fmicb.2017.00108/full

  • Also in Table 1, the example for ELISA is not very representative. One can easily find LODs for E.coli detection with ELISA below 10^3 CFU/g. https://pubs.rsc.org/en/content/articlelanding/2013/ay/c3ay40893k#!divAbstract
  • Following these comments: if we compare the performance (LOD) of the authors’ device with these analytical techniques, the authors’ device/solution performs worse. Although I reckon the advantages of a portable technique, the authors should try to compare their technique with other colorimetric bacterial detection methods. The performance of the proposed solution can be assessed through this more meaningful comparison. There are many methods for colorimetric bacterial detection – also many with better LODs. E.g. https://www.sciencedirect.com/science/article/pii/S0039914019300396
  • The current LOD of the device is far from the recommended standard of 5*10^2 CFU/ml for tap water – which is the biggest problem concerning the planned application. If the authors could give more examples (supported by papers) where this LOD could be applicable in practice with their device, it would improve the value of the paper.

Reviewer 2 Report

This study introduces a portable device for the detection of viable bacteria in water. The reagent used contains MTT and PMS which enables colorimetric detection. This approach is simple and rapid which shows tremendous potential for water quality measurement. The following comments should be addressed before the paper can be accepted for publication:

  • In abstract and conclusion, the authors mentioned that the proposed device is a “promising” or “successful” detection tool which is inappropriate. Since future works are needed to further improve the performance or sensitivity, the term “potential” should be used instead.
  • In introduction, the authors should briefly discuss the existing technologies for E. coli detection which include but are not limited to the followings:

- A microwave matrix sensor for multipoint label-free Escherichia coli detection (2020). Biosensors and Bioelectronics147, 111784.

- A microfluidic colorimetric biosensor for rapid detection of Escherichia coli O157: H7 using gold nanoparticle aggregation and smart phone imaging (2019). Biosensors and Bioelectronics124, 143-149.

- An integrated paper-based sample-to-answer biosensor for nucleic acid testing at the point of care (2016). Lab on a Chip16(3), 611-621.

  • The authors should discuss the advantages of the proposed device over the existing ones.
  • In introduction, it was mentioned “However, the above methods have many disadvantages including the following…. (1-5)”. The authors should discuss why is it so important to achieve the (1)-(5) criteria? Are authors suggesting that this technology can potentially be used in point-of-care or remote settings?
  • The authors mentioned that the cost of the proposed device is low. Please state the estimated cost in the text. Cost breakdown should be provided as well.
  • Fig 1: The schematic diagram should show a step-by-step procedure on how to detect bacteria from sample preparation to signal detection. The real image of device should be presented too.
  • Fig 3, 4a and 5a: data of control samples (0 bacteria) should be added to the graphs.
  • The authors should compare the detection limit/ sensitivity of the proposed device and the existing assays or technologies and suggest some potential solutions in details. How to improve the detection sensitivity?

Round 2

Reviewer 1 Report

I reckon that the authors undertook the effort to correct the mistakes in the manuscript and also by adding relevant data and comparison I think the relevance of the paper improved as well. Thus I now can support its publication in the journal.

Reviewer 2 Report

The authors have addressed my comments and I would recommend acceptance of this manuscript.